# Employing a Wireless Sensing Network for AIoT Based on a 5G Approach

**Sung-Jung Hsiao** 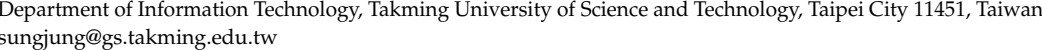

Department of Information Technology, Takming University of Science and Technology, Taipei City 11451, Taiwan;
sungjung@gs.takming.edu.tw

**Abstract:** In this paper, wireless sensing networks are considered in the context of the 5G communication architecture in order to ensure the transmission efficiency of the sensed data transmitted by the system. The various types of environmental sensors are very diverse. Wireless sensing networks may include many different types of sensing devices or data related to images or pattern transmission, among others, and are often limited by the problem of insufficient network bandwidth. By using 5G communications for data transmission, the problem of limited network bandwidth can be solved. In addition to the use of 5G transmission, when the NB-IoT method is used within the 5G network environment, it is much more efficient than that under the original LTE network conditions. Therefore, using 5G to transmit data provides the advantages of high transmission efficiency and data integrity. In this paper, in addition to analyzing the development of 5G technology, the proposed approach uses MATLAB software to simulate the generation of 5G signals under various parameter settings representing a range of conditions. Finally, our approach discusses the use of a 5G communication module, including driver installation and data transmission testing. At present, the 5G network architecture is still under construction worldwide. According to the transmission speed test data obtained in this study, the transmission efficiency of 5G is better than that of precursor generations.

**Keywords:** 5G; wireless sensor networks; NB-IoT; network slice

## 1. Introduction

The method proposed in this paper uses 5G communication technology for the transmission of sensed data. The various remote sensor devices available at present are increasingly diverse; the associated systems may also require that image data are transmitted in real-time. To address these issues, the wireless sensor network system proposed in this paper uses 5G communication for data transmission [1,2]. In Figure 1, the sensors of the system are placed at the far end. The system instantly transmits the remote environmental data to the embedded hardware module by means of 5G communication. The method proposed in this study utilizes various AI algorithms for the calculation and classification of the sensed data within embedded hardware modules. Then, through 5G transmission, the system transmits the processed information to a cloud database. Other remote operators can use various intelligent communication devices for remote monitoring and control. Of course, the system also has the capabilities of automatic judgment and analysis [3] and can immediately notify the operator to suggest how to deal with a given problem.

Please refer to Table 1 for the comparison table of abbreviations and full names of relevant communication terms in this paper.

The development of mobile communication technology based on cellular architectures has shifted from one-way (e.g., pagers) to two-way modes, simplex (e.g., walkie-talkie) to duplex modes, from analog modulation to digital modulation, from circuit switching to packet switching, and from pure voice services to data services. Moreover, the rapid development of multi-media services from low- to high-speed data services has realized the initial dream of mobile communication: anyone, at any time and any place, can talk to

anyone. In addition, it is also possible to initiate phone and video calls, access the internet, send and receive e-mails, use electronic services, upload and download files, or share photos and videos using high-speed mobile processes. In the future, it will not only be necessary to realize interconnection and communication between people and things, but also to enter the new communication era of the internet of things, in which all things are interconnected. Figure 2 intuitively shows us that, to date, mobile communication technology has experienced a revolutionary leap almost every 10 years [4].

**Table 1.** Abbreviated terms used in this paper are compared with full names.

| Abbreviation | Full Name |
|---|---|
| AIoT | Artificial Intelligence and Internet of Things |
| NB-IoT | Narrow Band-Internet of Things |
| 5G | 5th generation mobile networks |
| LTE | Long Term Evolution |
| WSN | Wireless sensor networks |
| IMT-Advanced | International Mobile Telecommunications- Advanced |
| TD-SCDMA | Time Division-Synchronous Code Division Multiple Access |
| WCDMA | Wideband Code Division Multiple Access |
| CDMA2000 | Code Division Multiple Access 2000 |
| TD-LTE | Time Division-Long Term Evolution |
| FDD-LTE | Frequency Division Duplexing-Long Term Evolution |
| EDGE | Enhanced Data rates for GSM Evolution |
| HSDPA | High Speed Downlink Packet Access |
| HSPA | High Speed Packet Access |
| WRC | World Radiocommunication Conference |
| HEW | High Efficiency Wireless |
| WLAN | Wireless Local Area Network |
| MTC | Machine Type Communication |
| MCC | Mission-critical control |
| KPI | Key Performance Indicator |
| RAN | radio access network |
| SDMA | Space division multiple access |
| 256QAM | Quadrature Amplitude Modulation |
| LDPC | Low-density parity-check code |
| DU | digital unit |
| BBU | baseband unit |
| RU | radio unit |
| MME | Mobility Management Entity |
| S/P-GW | SGW: Serving Gateway, PGW: Public Data Network Gateway |
| UNB | Ultra-Narrow Band |
| EC-GSM | Extended coverage-GSM |
| SGSN | Serving GPRS Support Node |
| S/P-GW | SGW: Serving Gateway, PGW: Public Data Network Gateway |

**Table 1.** *Cont.*

| Abbreviation | Full Name |
|---|---|
| UNB | Ultra-Narrow Band |
| EC-GSM | Extended coverage-GSM |
| SGSN | Serving GPRS Support Node |
| IMT-2000 | International Mobile Telecommunications-2000 |
| 3GPP | 3rd Generation Partnership Project |
| GSM | Global System for Mobile Communications |
| IS-95 | Interim Standard-95 |
| 3GPP2 | 3rd Generation Partnership Project 2 |
| *Cat.0* | Category 0 |
| LET-M | Long Term Evolution-Machine to Machine |
| LET-A | Long Term Evolution-Advanced |
| GPRS | General Packet Radio Service |
| MIMO | Multi-input multi-output |
| CA | Carrier aggregation |
| NFV | Network Function Virtualization |
| SDN | Soft defined network |
| ITU | International Telecommunication Union |
| URLLC | Ultra-reliable low-latency communication |
| AR | augmented reality |
| VR | virtual reality |
| eMBB | Enhanced mobile broadband |
| NR | New Radio |
| TDD | Time Division Duplexing |
| LAA | Licensed Assisted Access |
| eLAA | Enhanced Licensed Assisted Access |
| LWA | LTE-WLAN aggregation |
| NaaS | Network as a Service |
| PCRF | Policy and Charging Rules Function |
| C-RAN | Cloud Radio Access Network |
| LPWAN | Low-power wide-area network |
| LPWA | Low-power wide-area |
| CIoT | Cellular Internet of Things |
| EPC | Evolved Packet Core |
| RAT | Radio access technology |

The first phase of the 5G specification in 3GPP Release-15 is to accommodate early commercial deployments. The first 5G evolution standard, 3GPP Release 16, was completed on 3 July 2020. It mainly added 5G super uplink technology, improved ultra-reliable and low-latency communication and large-scale machine-type interconnection scenarios, and further improved energy efficiency and user experience. In December 2021, Qualcomm and MediaTek will have released baseband products that support Release16. 3GPP and that are expected to commercialize the 3GPP Release 18 (5G-Advanced) standard around 2025 and to provide a downlink rate of 20 Gbps and an uplink rate of 10 Gbps.

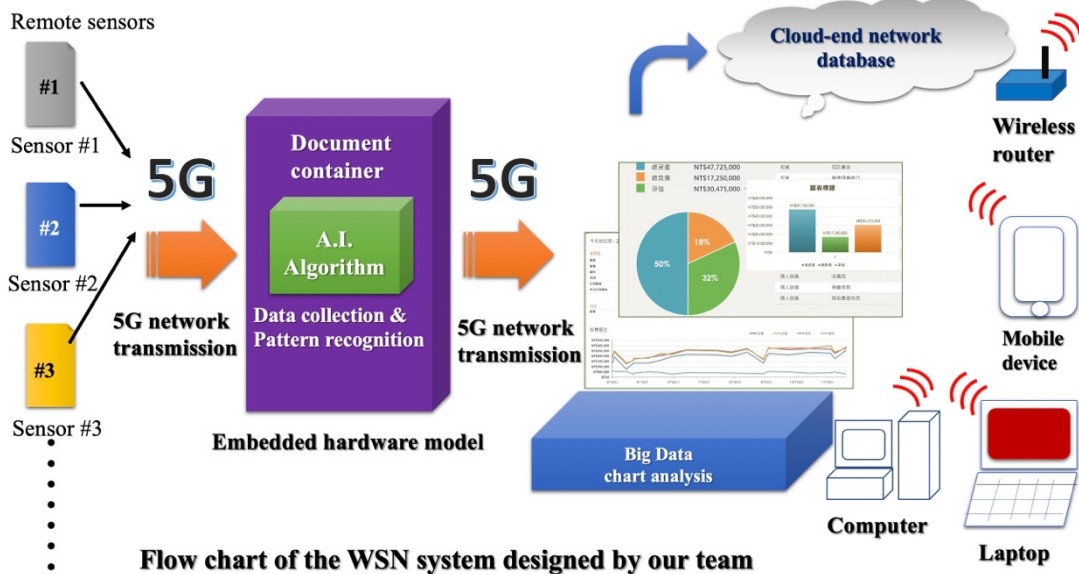

**Figure 1.** Wireless sensor networks using 5G communication transmission.

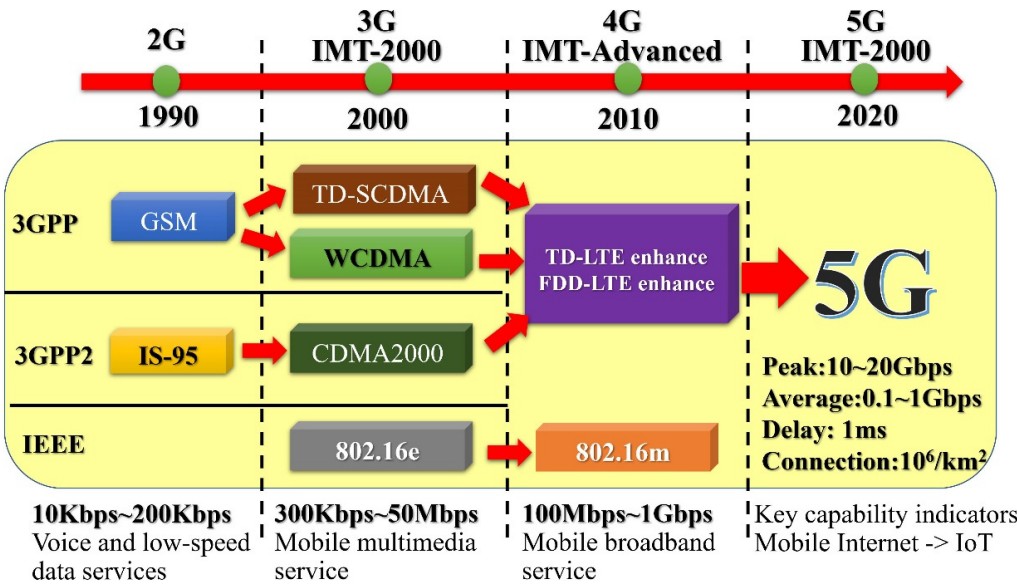

**Figure 2.** Mobile communication leaps every 10 years.

The 5G development momentum continues to increase. Although the threat of COVID-19 is still not far away, bringing many uncertainties to the market, global operators have not stopped in the commercial deployment of 5G. According to the latest development report on global 5G networks, spectrum and devices in the first half of 2021 released by the Global Mobile Equipment Suppliers Association (GSA) at the end of June 2021, the number of 5G device styles released worldwide had exceeded 800 for the first time by the end of May, reaching 822 models.

In terms of 5G network service development, the GSA report also shows that, as of the end of May 2021, a total of 443 telecom operators in 133 countries/regions had invested in 5G technology, and 169 operators had launched in more than 70 countries/regions one or more 5G network services compliant with 3GPP standards. What is more noteworthy is that the public network service of 5G independent network (SA)s ha recently entered large-scale commercial use from the test or trial stage, which is expected to further maximize the potential of 5G technology.

At this stage, the development of sub-6GHz 5G has matured, and related applications to business opportunities have gradually fermented; 5G SA, millimeter waves, open radio access network (Open RAN) architecture, private networks and other technology developments have been the focus of recent active efforts of players within the 5G ecosystem.

Currently, 5G SA is a prerequisite for the realization of ultra-reliable and low-latency communication (URLLC) and network slicing (Network Slicing). mmWave can provide wider bandwidth and release enhanced mobile broadband (eMBB) performance. The development of Open RAN brings new changes and development opportunities to the 5G ecosystem, and enterprise private networks are the focus market for the first wave of open RAN applications. In order to advance 5G from the current sub-6GHz to the above-mentioned new technology development stage, major technology manufacturers have stepped up investment in integrated access and backhaul (IAB), enhanced beam management, network slicing, and dynamic spectrum sharing (DSS), and various innovative technologies that can further improve the performance of 5G base stations, terminal devices, and network services.

In fact, in the 4G era, the developmental and evolutionary path of mobile communication networks had two branches, covering more application scenarios, as shown in Figure 3. One of the branches is the broadband era of large traffic volumes, high speeds, and high-speed mobile technology, while the other is the era of the internet of things, featuring small data, wide coverage, and large capacity [5].

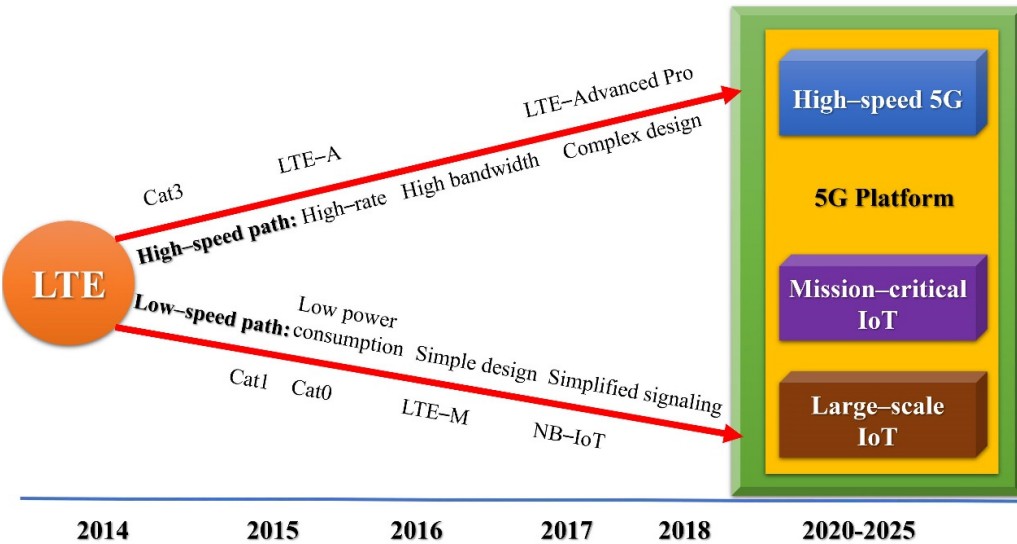

**Figure 3.** Evolutionary branches of mobile communication.

Therefore, 5G mobile communication technology is expected to be necessary to meet the huge growth in the number of mobile communication users in the future (i.e., to increase the network capacity), as well as to meet the huge business needs of the internet of things and the ultra-high-speed data transmission rate requirements. Moreover, in addition to the evolution of the mobile communication network architecture (i.e., the so-called fifth generation), 5G mobile communication technology comprises evolution in three key dimensions, as shown in Figure 4.

Due to the widespread adoption of the following two technologies, the spectrum utilization efficiency of mobile communications has also evolved and improved [6,7].

1. Advanced modulation technology: QPSK→16QAM→64QAM→256QAM; and
2. Multi-antenna technology: MIMO2 × 2 → MIMO4 × 4 → MIMO8 × 8 → MIMO64 × 64 → Massive MIMO256 (large-scale smart antenna array).

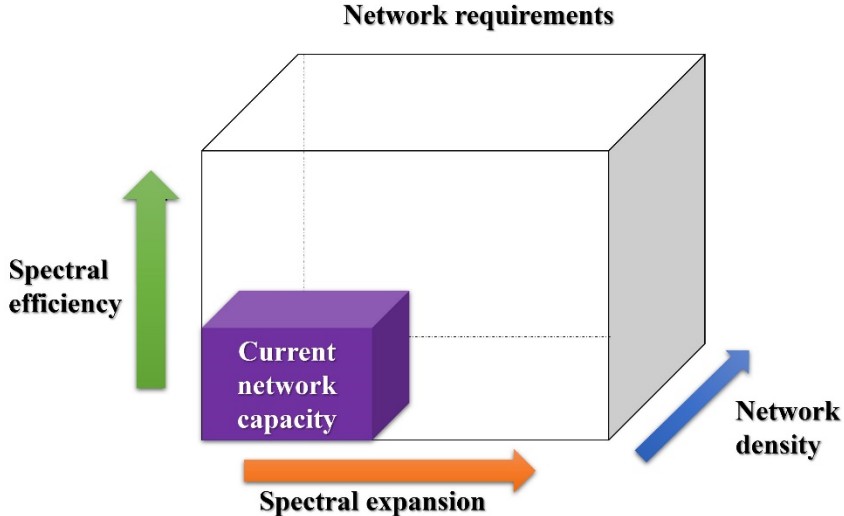

**Figure 4.** Evolution of mobile communication technologies.

The evolution of spectral efficiency in mobile communications is shown in Figure 5.

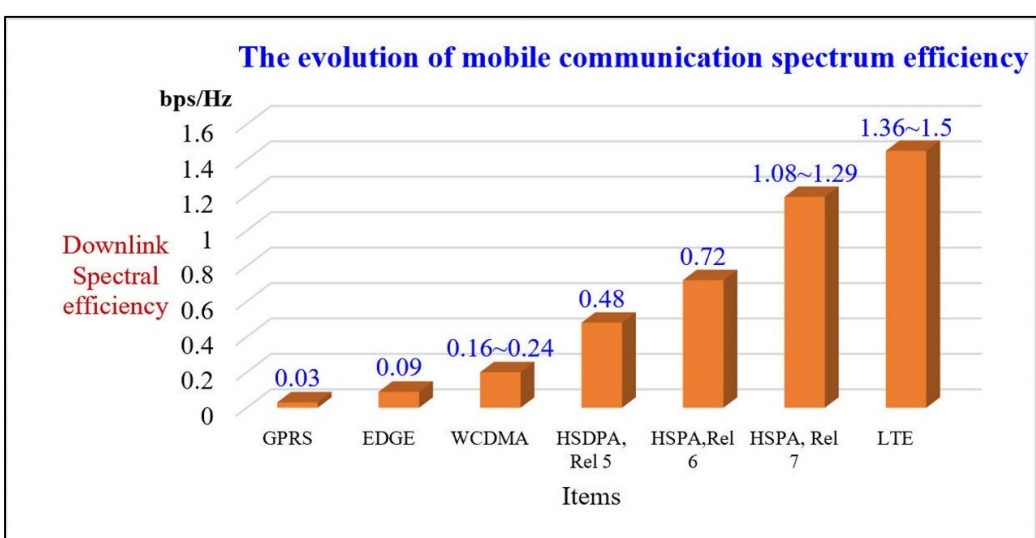

**Figure 5.** Evolution of mobile communication spectral efficiency.

Spectrum utilization is coming closer and closer to the limit of Shannon's law. In addition, in order to support ultra-high data transmission rates, in addition to the above-mentioned improvement in spectral efficiency, the operating bandwidth of a cell will naturally become larger and larger, or multi-carrier aggregation (CA) technology can be introduced to improve the cell bandwidth, as follows:

30 kHz→200 kHz→1.25 MHz→5 MHz→10 MHz→20 MHz→100 MHz→200 MHz.

The wireless frequency band has also been expanding to higher frequencies, up to the millimeter wave band (>20 GHz), as follows:

700 MHz→900 MHz→1800 MHz→2100 MHz→2600 MHz→3 GHz→6 GHz→10 GHz→30 GHz.

The higher the frequency of radio waves, the greater the propagation loss and the worse the penetration, resulting in lower, weaker cell coverage; see Figure 6.

The higher the frequency band, the smaller the cell, and the higher the density of the natural peak-to-nest network, such that more operators need to deploy more base stations. In the future, 5G base stations are expected to be widely distributed; for example, every lamppost along a street may include a base station. There may be one public base station installed, with a private base station also installed in every home. Faced with the huge

scale of such network stations, the cost of mobile communication network construction and maintenance are also expected to increase [8,9]. Therefore, future 5G mobile communication networks must be flexible enough that they have strong autonomy, adaptability, and creativity, as well as the ability to learn from the wireless environment. The base stations must work co-operatively with each other, self-adaptively optimizing and configuring to achieve high reliability and high-speed communication anytime and anywhere. Moreover, they should efficiently utilize the limited wireless spectrum resources in a heterogeneous network environment. Only in this way can network function virtualization (NFV), collaboration, the cloud, and the soft-defined network (SDN) be realized, while keeping network maintenance costs sufficiently low.

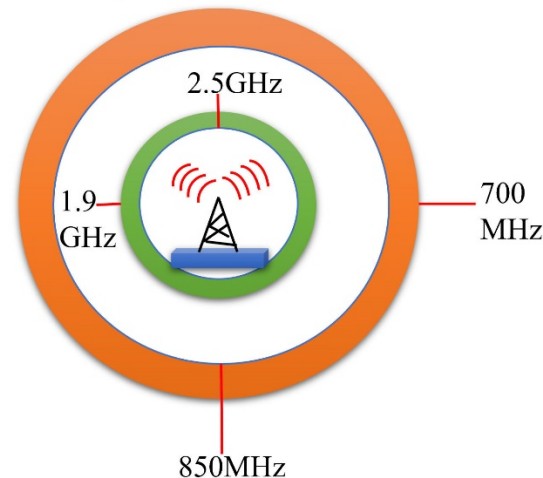

**Figure 6.** Coverage of cells in different mobile communication frequency bands.

## 2. Fifth Generation (5G) Application Scenarios

In 2016, the International Telecommunication Union (ITU) officially named 5G in the IMT-2020. Figure 7 shows the 5G standardization progress schedule of standards organizations, such as 3GPP [10].

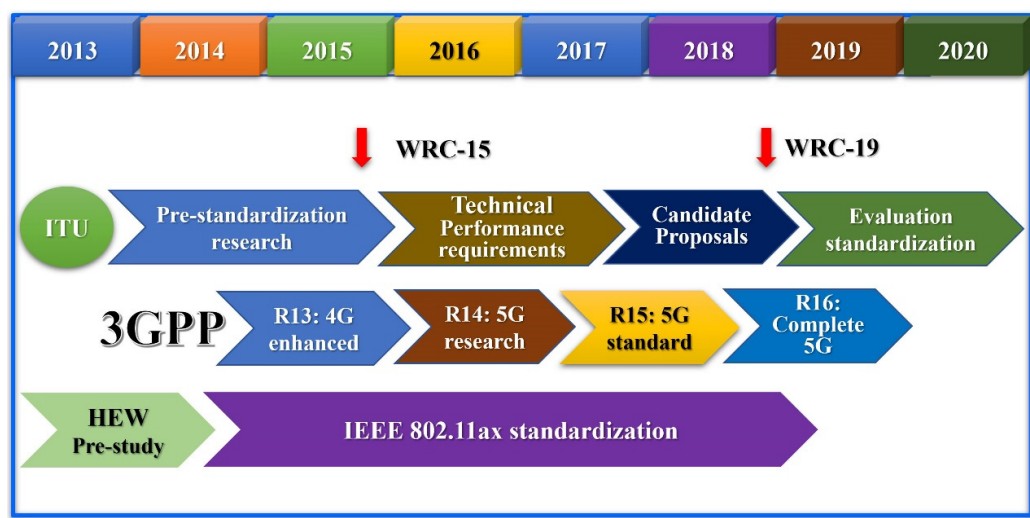

**Figure 7.** Fifth generation (5G) standardization timeline.

Our approach can categorize the 5G standardization schedule into three styles:

(1)  ITU

At present, the 5G vision research has been completed: the collection of 5G technical solutions was launched at the end of 2017, and the formulation of 5G standards was completed in 2020.

(2)  The Third Generation Partnership Project (3GPP)

In early 2016, 5G standards research began. The first version of the 5G standard was formulated and completed in the second half of 2018, and the full version of the 5G standard meeting the ITU requirements was completed by the end of 2019.

(3)  IEEE

The next-generation WLAN (802.11ax) standard formulation was initiated in early 2014, and the standard formulation was expected to be completed in early 2019.

However, the latest news is that the progress of the 5G standard led by 3GPP needs to be accelerated, and the first version of the standard R15 was expected to be released in the first half of 2018, half a year ahead of schedule [11,12].

The following three application scenarios are usually included in 5G (please refer to Figure 8):

1.  Massive internet of things (massive IoT/MTC/M2M)

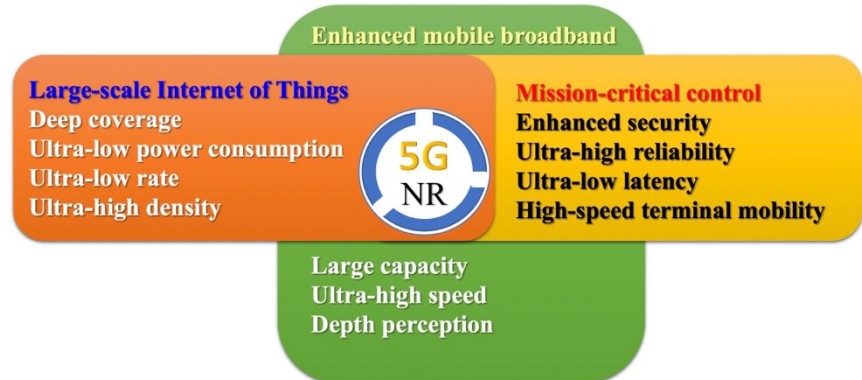

**Figure 8.** Three major application scenarios of 5G.

The massive internet of things is characterized by a large number of connected devices (ultra-high density), ultra-low power consumption, deep coverage, and ultra-low complexity, and includes applications such as remote meter reading and logistics tracking management.

2.  Mission-critical control (MCC)

Mission-critical IoT is mainly used in areas such as unmanned driving, automatic factories, and smart grids, requiring ultra-high security, ultra-low latency, and ultra-high reliability. It is also known as ultra-reliable low-latency communication (URLLC). For example, when our research wants to experience services such as augmented reality (AR) or virtual reality (VR), remote control, and remote game consoles, the data need to be sent to the cloud for analysis and processing, while the processed data or instructions must be sent back in real-time. The delay of this round-trip process must be low enough such that the user cannot perceive it. In addition, machines are more sensitive to delay than humans and have higher requirements for delay, especially in 5G applications such as the internet of vehicles, automatic factories, remote robots, and telemedicine and remote robotic surgery.

3.  Enhanced mobile broadband (eMBB)

With ultra-high transmission rates (>10 Gbps), the 5G era will be oriented towards applications such as 4K and 8K ultra-high-definition video, holographic technology, augmented/virtual reality, and so on. The main demand for mobile broadband is higher data transmission rates.

This paper mainly considers large-scale 5G internet of things technology; in particular, narrowband internet of things (NB-IoT).

## 3. Fifth Generation (5G) Core Indicators

Two core indicators have been clearly defined in the 5G New Radio (NR) standard:

1.  A peak rate of DL 20 Gbp; and
2.  A user plane delay of 0.5 ms (URLLC).

These two key KPI values have increased by 20 times, compared to those in 4G LTE. Table 2 provides details of the 5G NR performance requirements.

**Table 2.** Fifth generation (5G) New radio interface KPI indicators.

| 5G key Indicator Project | 5G KPI Items | KPI Value |
|---|---|---|
| Peak rate | Peak data rate | DL: 20 Gbps<br>UL: 10 Gbps |
| Peak spectral efficiency | Peak spectral efficiency | DL: 30 bps/Hz<br>UL: 15 bps/Hz |
| Control plane delay | Control plane latency | 10 ms |
| User plane delay | User plane latency | URLLC: 0.5 ms (DL&UL) |
| Infrequent small packet delay | Latency for infrequent small packets | TBD |
| Mobility interruption latency | Mobility interruption time | 0 ms |
| Inter-system mobility | Inter-system mobility | Mandatory/Optional |
| Reliability | Reliability | URLLC: BLER $\leq$ 0.01% (1 ms) |
| Cover | Coverage | mMTC: 164 dB |

This section focuses on the eight key technologies that 5G may adopt, including new technologies involved in both the radio access network (RAN) and network architecture [13].

### 3.1. mmWave Technology

In the past, the traditional working frequency bands of mobile communication have mainly been concentrated below 3 GHz, making the spectrum resources very crowded. However, in the high-frequency bands (e.g., millimeter wave, centimeter wave, and mmWave bands), the available spectrum resources are abundant, which can effectively alleviate the current situation of tight spectrum resources. This can help to realize extremely high-speed, short-distance communications, thus, supporting the needs of large capacity and high speed in 5G technologies [14].

The application of high-frequency bands in mobile communications is the future development trend, with which the industry is highly concerned. The main advantages of high-band mmWave mobile communications are listed below:

1.  Enough available bandwidth;
2.  Miniaturized antennas and devices;
3.  Higher antenna gain;
4.  Good diffraction ability; and
5.  Suitability for deploying large-scale antenna arrays (massive MIMO).

However, high-frequency millimeter wave mobile communications also have shortcomings, such as a short transmission distance, poor penetration ability, and being easily affected by environmental factors. The problems related to RF devices and system design also need to be further studied and addressed.

At present, major research institutions and companies are actively conducting research on high-frequency band requirements and the selection of potential candidate frequency bands. Although high-frequency band resources are relatively abundant at present, scientific planning and overall consideration are still needed to optimize the allocation of valuable spectrum resources [15].

### 3.2. Massive Antenna Array

Multi-antenna technology has undergone development from passive to active, from two-dimensional (2D) to three-dimensional (3D), from high-order MIMO to massive array (massive MIMO), and it is expected to achieve dozens of times (or even higher) spectral efficiency improvements, comprising one of the most important 5G technology research directions.

Due to the introduction of active antenna arrays and millimeter wave technology, the number of co-operative antennas that can be supported by the same physical space on the base station side is expected to reach 128 or more, as shown in Figure 9. In addition, the original 2D antenna arrays are expanded into 3D antenna arrays, forming novel 3D MIMO or stereo multi-dimensional MIMO technology, thus, supporting intelligent multi-user beam shaping and reducing interference between users. Combined with high-frequency millimeter wave technology, this is expected to further improve wireless signals and override performance. The 3D MIMO technology adds a vertical dimension to the original MIMO, such that the beams are three-dimensionally shaped in space, better avoiding mutual interference. With massive MIMO, multi-directional beamforming can be achieved.

#### Physical plane of 20 cm × 20 cm antenna

| Antenna element Spacing($d$) | 3.5 GHz ($\lambda = 8.6\ cm$) | 10 GHz ($\lambda = 3\ cm$) | 20 GHz ($\lambda = 1.5\ cm$) |
|---|---|---|---|
| 0.5 $\lambda$ | 16 | 169 | 676 |
| 0.7 $\lambda$ | 9 | 81 | 361 |

**Figure 9.** Schematic diagram of the principle of massive MIMO.

At present, researchers are conducting research on large-scale antenna channel measurement and modeling, array design and calibration, pilot channels, codebooks, and feedback mechanisms. In the future, more users will be supported by space division multiple access (SDMA), which can significantly reduce the transmission power, achieve green energy savings, and improve coverage.

### 3.3. New Modulation and Coding Technology

Modulation and coding technology is the core technology and crown jewel of mobile communication. The new modulation and coding technologies adopted in 5G mainly include 256QAM high-order modulation, LDPC, and polar coding and decoding technologies. They are introduced separately below.

In 1948, Shannon first proposed a method for achieving reliable communications in noisy channels, in his seminal paper Mathematical Theory in Communication, where he proposed the famous disturbed channel coding theorem, laying the foundation for error correction and the basics of coding.

In the early 1950s, Hamming, Slepian, Prange, and others designed a series of encoding and decoding schemes with excellent performance based on Shannon's theory (i.e., the limit of various channels under coded channel conditions). As the performance limit of communication systems, the Shannon limit is of great significance; it also drives the research into and application of error correction codes designed and constructed to approach the Shannon limit in the field of communications.

Simply put, channel coding involves redundant bits being inserted into a $k$-bit data block to form a longer code block. The length of this code block is $n$ bits ($n > k$), and $n - k$ bits are used for detection and error correction. The redundant bits of the coding rate (R) are given by $k/n$. A good channel coding is achieved when, at a certain coding rate, it can be infinitely close to the theoretical limit of the channel capacity; that is, the Shannon limit.

The determinants of 3GPP deciding which encoding method to use for 5G include decoding throughput, delay, error correction capability, block error rate (BLER), and flexibility, as well as the complexity, maturity, and backward-compatibility of the associated hardware and software implementations, among other factors.

The low-density parity check (LDPC) code, first proposed by Dr. Robert G. Gallager of MIT in 1963, is a kind of linear block code with a sparse check matrix. It not only has good performance that is close to the Shannon limit, but also has low decoding complexity and flexible structure, which have always been research hotspots in the field of channel coding.

*3.4. Multi-Carrier Aggregation*

LTE R12 already supports the aggregation of five 20 MHz carriers, as shown in Figure 10. On the other hand, 5G will expand this to support the aggregation of up to 32 carriers. In addition, future 5G networks will be converged networks, and carrier aggregation technology will be greatly extended to support carrier aggregation between the following various types of wireless links (as shown in Figure 11): The aggregation of up to 32 carriers within LTE, carrier aggregation between systems and 3G-HSPA, and radio links. The system supports FDD and TDD link aggregation (i.e., uplink and downlink asymmetric carrier aggregation), and supports LTE licensed spectrum-assisted access (LAA/eLAA), which supports carrier aggregation with unlicensed spectra, such as Wi-Fi wireless links.

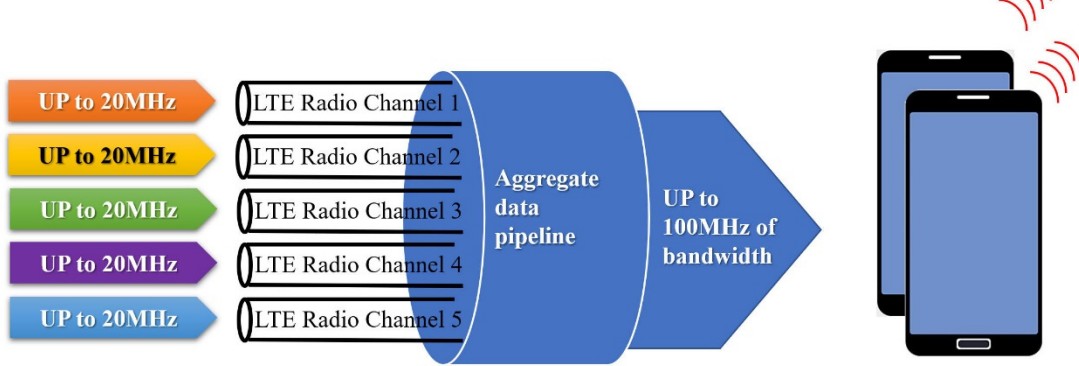

**Figure 10.** Schematic diagram of the five-carrier aggregation in LTE.

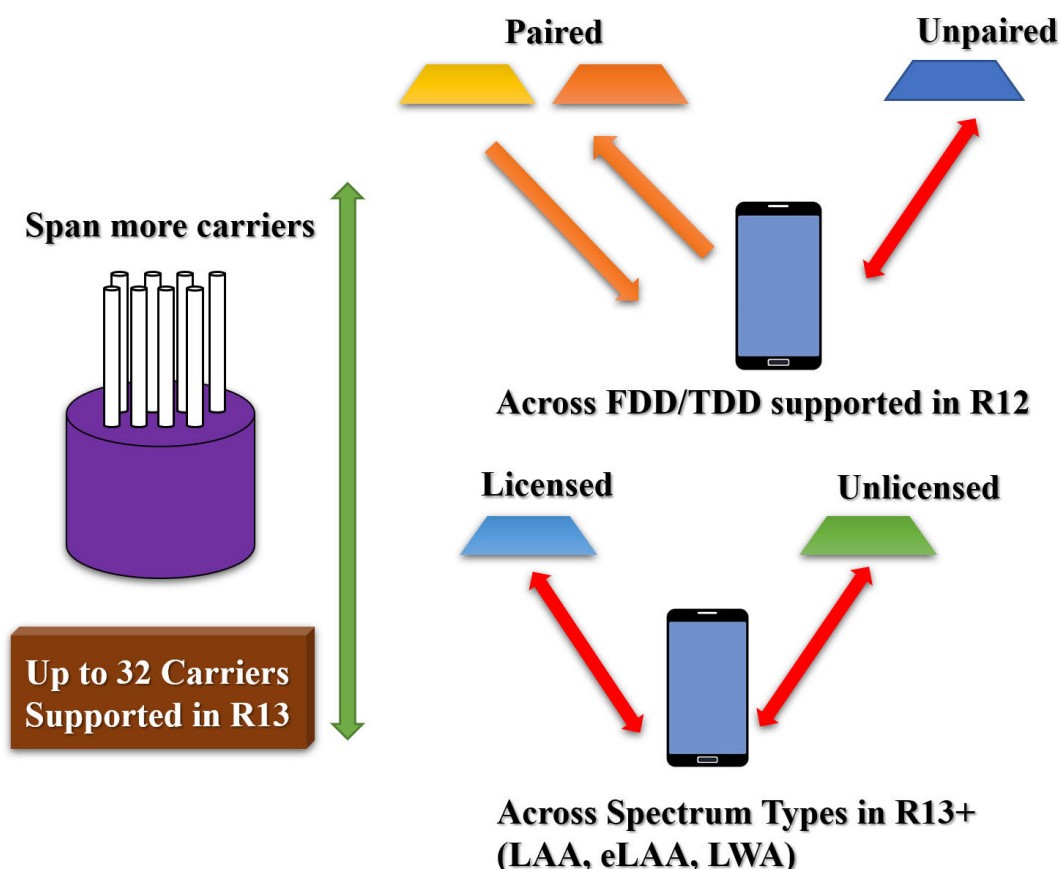

**Figure 11.** Fifth generation (5G) technology supports carrier aggregation technology between multiple radio links.

### 3.5. Network Slicing Technology

The simplest explanation of network slicing technology is that a physical network can be divided into multiple virtual end-to-end networks. The core network and the virtual network are logically independent; the failure of any one virtual network will not affect any other virtual network. Each virtual network is similar to the pliers and saw on a Swiss army knife; they have different functions and characteristics and are oriented to different needs and services. They can be flexibly configured and adjusted, and users can even customize the network functions and services to achieve network as a service (NaaS).

At present, the main terminal equipment in the 4G network is the mobile phone, the wireless access network part of the network—including the digital unit (DU) or the baseband unit (BBU) and the radio unit (RU)—and the core network part, which adopts special equipment provided by equipment manufacturers.

The 4G network mainly serves people, and the main devices connected to the network are smartphones, which do not require network slicing for different application scenarios. However, 5G networks need to divide the physical network into multiple virtual logical networks, each of which corresponds to a different application scenario; this is called network slicing. A summary of 5G network slicing technology is shown in Figure 12.

In order to realize network slicing, the pre-requisite for the system is network function virtualization (NFV). In essence, NFV is used to transfer the software and hardware functions of specialized equipment in the network—such as MME, S/P-GW, and PCRF in the core network, the digital unit (DU) in the wireless access network, and so on—to the virtual network. These web hosts (virtual machines, VMs) are based on industry standard commodity servers that are low-cost and easy to install. Simply put, they are intended to replace the dedicated network element equipment in the network with industry standard

servers, in order to realize decoupling of the hardware and software from the network equipment and to achieve rapid development and deployment.

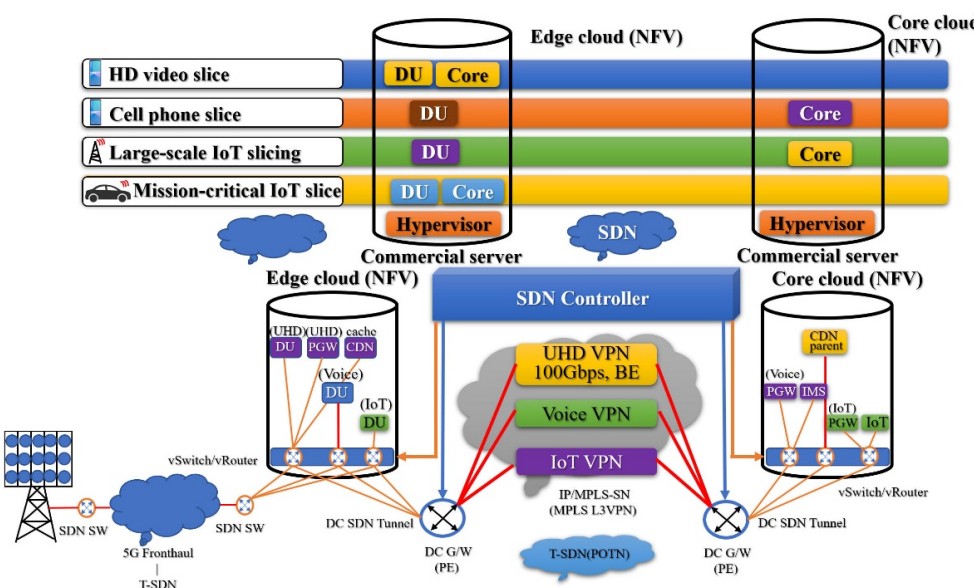

**Figure 12.** A diagram of 5G network slicing technology.

After the network is functionally virtualized, the radio access network part is called the edge cloud, and the core network part is called the core cloud. The VMs in the edge cloud and those in the core cloud are interconnected through SDN (software-defined networking), which also realizes decoupling of the hardware and software of the network devices and achieves complete separation of the controller and bearer.

As shown in Figure 12, under different application scenarios, the network can be divided into four slices:

1. Ultra-HD video slice

Originally, after the digital unit (DU) and some core network functions in the network are virtualized, storage servers are added, which are unified into the edge cloud. Moreover, some virtualized core network functions are put into the core cloud.

2. Phone slice

The digital unit (DU) of the wireless access part of the original network is virtualized and put into the edge cloud. The core network functions of the original network, including IMS, are virtualized and put into the core cloud.

3. Massive IoT slice

As most of the sensors are stationary and do not require mobility management, the core cloud's task is relatively simple in this slice.

4. Mission-critical IoT slice

Due to the high latency requirement, in order to minimize the end-to-end latency, the core network functions and related servers of the original network, are all moved to the edge cloud.

Of course, network slicing technology is not limited to these types of slices. It is flexible, and operators can customize their own virtual networks according to the application scenario.

### 3.6. Ultra-Dense Heterogeneous Network

In future 5G communications, wireless communication networks are expected to evolve in the direction of network diversification, broadbandization, integration, and intelligence. With the popularization of various smart terminals, data traffic will see intense growth. In the future, data services will mainly be distributed in indoor and hotspot areas,

making ultra-dense heterogeneous networks one of the main means to achieve the 1000-fold capacity demand of 5G technologies.

The 5G network will adopt a three-dimensional, layered, and ultra-dense heterogeneous network, and a variety of cells will be deployed in the macro cell network layer, including many micro-cells, pico-cells, and femto-cells, with coverage ranging from a few hundred meters to a dozen meters. Ultra-dense networks can improve network coverage, greatly increase the system capacity, and allow for the offloading of services, enabling more flexible network deployment and more efficient frequency reuse. In the future, for high frequency bands and large bandwidths, more dense network solutions should be adopted, and more than 100 small cells/sectors may be deployed.

At the same time, the increasingly dense network deployment also makes the network topology more complex. Inter-cell interference has become the main factor restricting system capacity growth, which greatly reduces the energy efficiency of networks. Interference elimination, rapid cell discovery, dense inter-cell co-operation, dynamic load balancing, and mobility enhancement solutions based on terminal capability enhancement are all current research hotspots in dense networks.

## 4. New Network Architecture

### 4.1. C-RAN

At present, the LTE access network adopts a flat network architecture, which reduces the system delay, as well as the network's construction and maintenance costs. In the future, 5G may adopt a cloud access network architecture, called Cloud-RAN (C-RAN).

C-RAN is a green radio access network architecture based on centralized processing, co-operative radio, and a real-time cloud computing architecture. The basic idea of C-RAN is to directly transmit wireless signals between remote antennas and centralized central nodes by making full use of low-cost, high-speed optical transmission networks, in order to construct service areas for wireless access systems covering hundreds of base stations, even spanning hundreds of square kilometers.

The C-RAN architecture is suitable for the use of collaborative technology, which can reduce interference, reduce power consumption, and improve spectral efficiency. At the same time, it is convenient to realize intelligent networking for dynamic use. Centralized processing is beneficial to reduce costs, facilitate maintenance, and reduce operating expenses. The current research contents include the architecture and functions of C-RAN, such as centralized control, the RRU interface definition of the baseband pool, and closer co-operation based on C-RAN, including elements such as the base station cluster, virtual cells, and so on.

### 4.2. SDN and NFV

The 5G network architecture is also expected to fully adopt SDN and NFV technologies. The increasingly mature and successful application of cloud virtualization technology in the IT industry and the open thinking of the internet have jointly driven major operators to rethink the architecture of and service deployment in mobile communication networks.

The concept of software-defined networking (SDN) involves allowing software to control the network and fully open network capabilities (i.e., new network architectures and network technology with three-dimensional characteristics). By introducing the concept of SDN, the traditional telecommunication network architectures, which are closed and vertically integrated, can be transformed into layered network architectures that are flexible, open, highly integrated, service-oriented, and that can ensure service quality.

## 5. Internet of Things: Features and Applications

### 5.1. IoT Features

Modern IoT is typically divided into three layers, as shown in Figure 13.

1.  Perception layer: The perception layer is responsible for collecting a large amount of information, through the use of sensors and terminal IoT chips.

2. Network layer: The network layer provides a safe and reliable connection, facilitates interaction and sharing, and is responsible for transmitting a large amount of information data collected by the perception layer to the application layer or third-party cloud for analysis and processing. Then, it sends back instructions and other related information from the terminal.

3. Management and application layer: The application layer analyzes big data and provides an open cloud service platform for third parties to make appropriate business decisions and provide services.

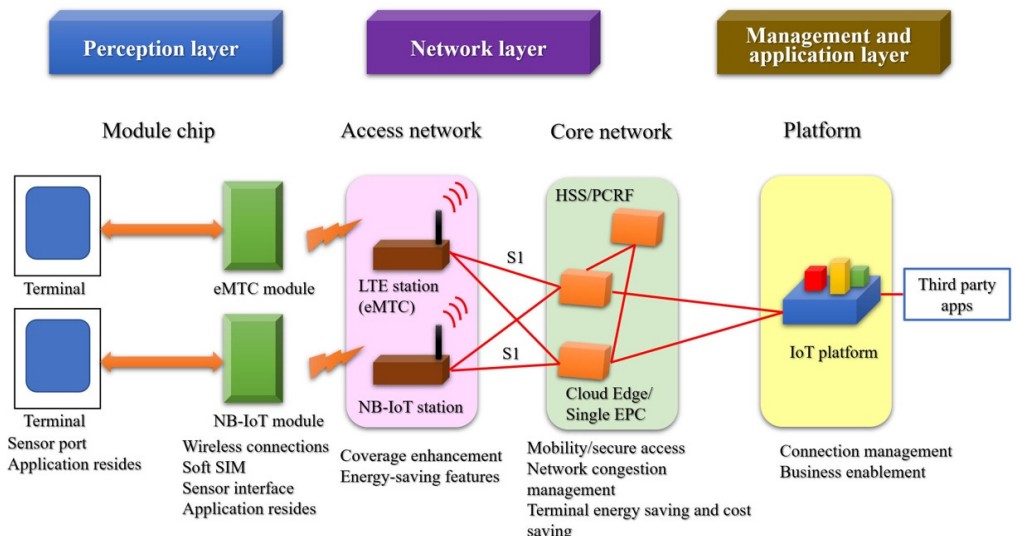

**Figure 13.** IoT technology architecture.

Generally, the internet of things has the following characteristics and requirements, as shown in Figure 14.

1. Super strong coverage: The coverage is enhanced by 20 dB, reaching MCL = 164 dB.
2. Ultra-high capacity: It supports large-scale connections.
3. Ultra-low power consumption: It has a 10-year battery life.
4. Ultra-low cost: It costs $5–10/terminal.
5. Lower rate: The rate is 10–100 kbps.
6. Delay tolerance: Its delay tolerance is 1–10 s.

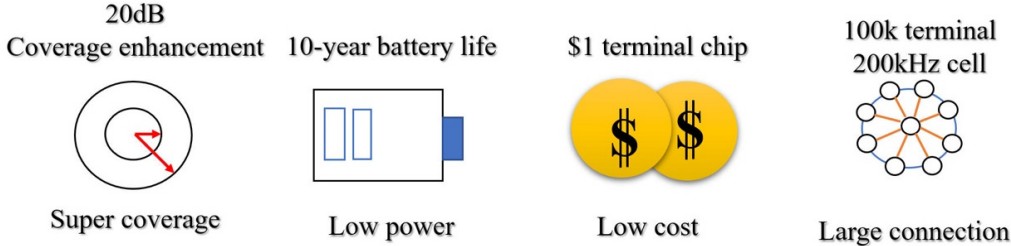

**Figure 14.** IoT feature requirements.

Low-power wide-area network (LPWAN) IoT has two key points:

1. Low power consumption; and
2. Wide area coverage.

Simply put, LPWAN IoT technology is a wireless communication network technology that achieves long-distance and deep coverage under the condition of reducing energy consumption.

### 5.2. IoT Applications

IoT applications can be divided into three categories, according to the requirements of speed, delay, and reliability.

1.  Low-latency, high-reliability services. This type of service has high requirements, in terms of throughput, delay, and/or reliability, and its typical applications include the internet of vehicles, telemedicine, and so on.
2.  Medium-demand business. This type of business has medium or low throughput requirements, where some applications have mobility and voice requirements, as well as certain restrictions on coverage and cost. Typical businesses include smart home defense and wearable devices.
3.  Low-power wide-area (LPWA) services. The main features of LPWA services include low power consumption, low cost, low throughput, the need for wide (deep) coverage, and high capacity. Typical applications include meter reading, environmental monitoring, logistics, and asset tracking.

### 5.3. IoT Technology Classification

Among the various IoT application services, low-power wide-area (LPWA) IoT services are the main market where operators around the world compete for connectivity, due to their large-scale connection requirements. At present, there exist a variety of IoT communication technologies that can carry LPWA services, such as GPRS, LTE, LoRa, Sigfox, and so on, which are classified and introduced below.

IoT technologies can be divided into the following two categories, based on the type of spectrum used (as shown in Figure 15):

1.  IoT technologies that use licensed spectra, such as EC (extended coverage)-GSM, NB-IoT, and LTE-M, are mainly constructed and operated by 3GPP-led operators and telecom equipment vendors, which can also be called the cellular internet of things (CIoT). The IoT technology classification of licensed spectra is shown in Figure 16.
2.  IoT technologies that use unlicensed spectra, such as LoRaWAN, Sigfox, Weightless, HaLow, Random Phase Multiple Access (RPMA), and other proprietary technologies, most of which are utilized in non-telecommunication fields.

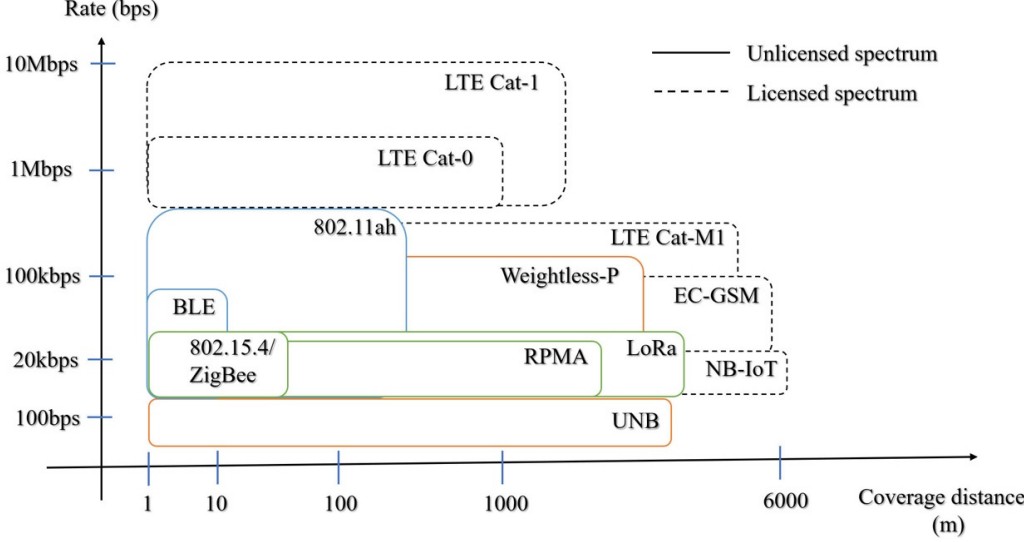

**Figure 15.** IoT Technology Classification.

Considering the coverage distance, IoT technology can be divided according to long-distance coverage cellular networks and short-distance non-cellular networks:

1.  Long-distance coverage (>1000 m): NB-IoT, Sigfox, LoRa.

2. Short-range coverage (<100 m): Wi-Fi, Bluetooth, NFC, ZigBee. These are suitable for device-to-device (D2D) direct communication in non-networking situations.

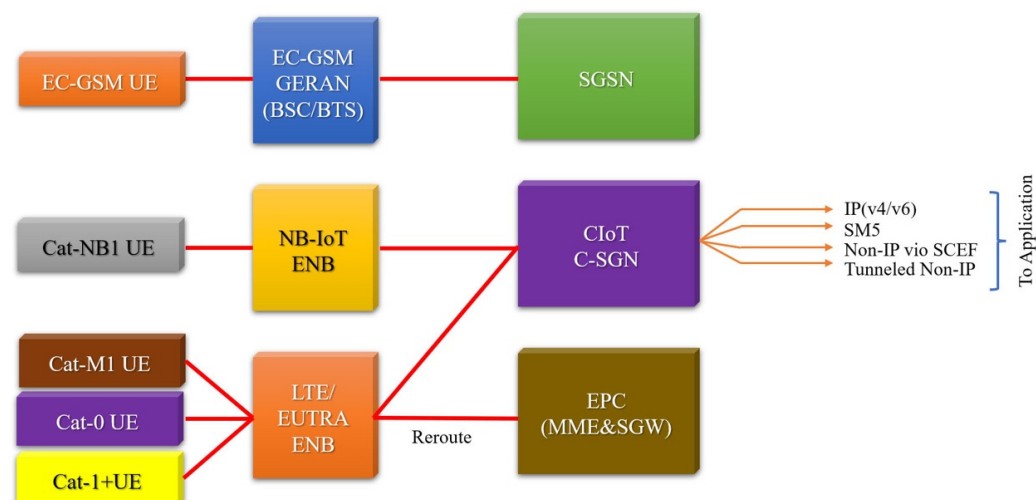

**Figure 16.** IoT Technology Classification of Licensed Spectra.

## 6. The Application of 5G Network Slicing Technology

At present, 5G technology is the most important mobile communication technology. Based on meeting the requirements of 5G systems, 5G network slicing can further facilitate the in-depth scheduling of communication. In the virtual network, 5G network slicing can be completed by use of a mapping relationship. Mapping refers to maintaining the current underlying physical network state and accurately matching physical nodes and links in physical resources under a specific network environment. However, existing mapping algorithms only consider system latency or operating efficiency unilaterally, while ignoring the parallel relationship between the two.

A network slicing system consists of a wireless virtual access network slice layer, a virtual operator layer, and an infrastructure provider layer. Virtual operators propose a virtual network router (VNR) and distribute the VNR for the virtualized management of network functions. Through the control of NFV MANO, the infrastructure provider realizes the VNFC configuration and deployment based on different VNRs, forms the internal baseband of different virtualized processing units, and connects to the corresponding virtualized remote radio frequency units, in order to realize the virtualized operator-based VNR requirements of different wireless virtual access network slice layers.

Suppose our method has a weighted undirected graph of a physical network $C = (Ai, Si)$, where $A_i = \{a_1, a_2, \ldots, a_n\}$ represents a set of network nodes, $S_i = \{s_1, s_2, \ldots, s_n\}$ is the computational levels for $A_i$, $L_n = \{l_1, l_2, \ldots, l_n\}$ is a link set composed of nodes, $l_n$ are the underlying specific physical links, the inefficiency of the physical network communication link is expressed as $\gamma_i$, and the bandwidth is $d_i$.

Combining queuing theory and network slicing technology to establish a two-level dynamic scheduling model of NFV MANO and NFVs in the physical network slicing system, the VNR of the network slicing system can be back-logged and stabilized within a certain range.

According to queuing theory, dynamic scheduling occurs twice. The first-level queuing dynamic scheduling process occurs at the system NFV MANO, where the queuing state transition function is:

$$Z(X) = \gamma_i + (A_i S_i - L_n) d_i \tag{1}$$

The second-level queuing dynamic scheduling process occurs at the NFVs, where the queuing state transition function is:

$$Z(Y) = \frac{\gamma_i + A_i L_n d_i}{Z(X)} \tag{2}$$

Based on the above analysis, the mapping model in the 5G network slicing architecture can be further obtained, which is expressed as:

$$F(x) = \frac{Z(X) + Z(Y)}{d_i} \gamma_i \tag{3}$$

In a network virtualization scenario, network requests for shards cannot always arrive one-by-one at a certain interval. According to the traffic characteristics of network slices, online mapping technology is used to realize dynamic allocation of the underlying physical resources, as well as scheduling them in the time dimension. As shown in Figure 17, incoming network slice requests are processed dynamically using time windows; that is, time is divided into a series of consecutive time windows. The network slices are processed within a specific time window, according to their priority order in the life cycle. The shorter the life cycle, the higher the priority processing level. According to the definition of reliability, the shorter the mapping life cycle, the lower the failure rate, which thereby improves the reliability of the entire network.

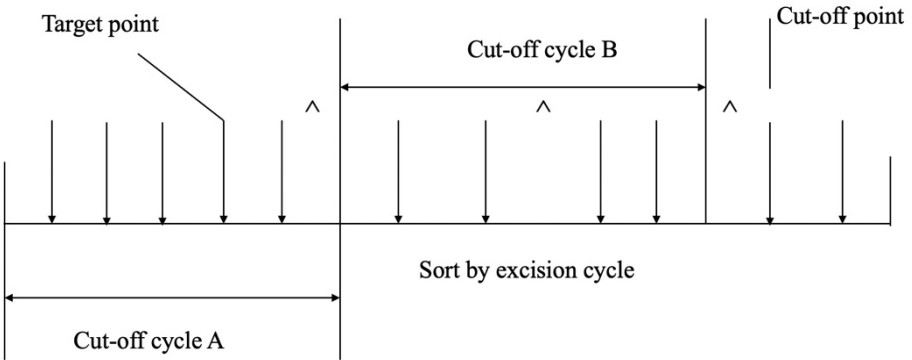

**Figure 17.** Network slice mapping mechanism on the time window line.

In order to verify the feasibility of the system, the MATLAB simulation tool was used to simulate and analyze the online mapping algorithm for the 5G network slices. MATLAB was used to simulate various 5G signals, as shown in Figure 18. Within the software, researchers can choose from different 5G waveform types. Furthermore, down-link and up-link options allow the operator to fully customize the content of the waveform. Researchers can also generate projects such as NR-TM, down-link FRC, and up-link FRC.

The simulation physical network generates ten physical nodes randomly, and the distribution interval for computing resources and link resources was 50 and 80, respectively, while the time units of different slice life cycles were 60, 80, and 100. A Poisson distribution was used to ensure that slices arrive randomly. The average probability of arrival rate of a single time window (50 time units) was 1.4. Using the Poisson distribution, the service time interval of each network slice is 15 time units. The computing resources and link bandwidths of the virtual nodes in the network system were randomly distributed within the interval [30, 50]. The RRU adopted a service rate mode of 1.1–1.4, the control parameter was within 0.2–0.6, and the analog period was set to 400 time units.

In the content of this section, this paper mentions an introduction to the use of 5G in MATLAB software. In the Appendix A, there is a more detailed 5G oper-ation of MATLAB, which is provided for reference by other related researchers.

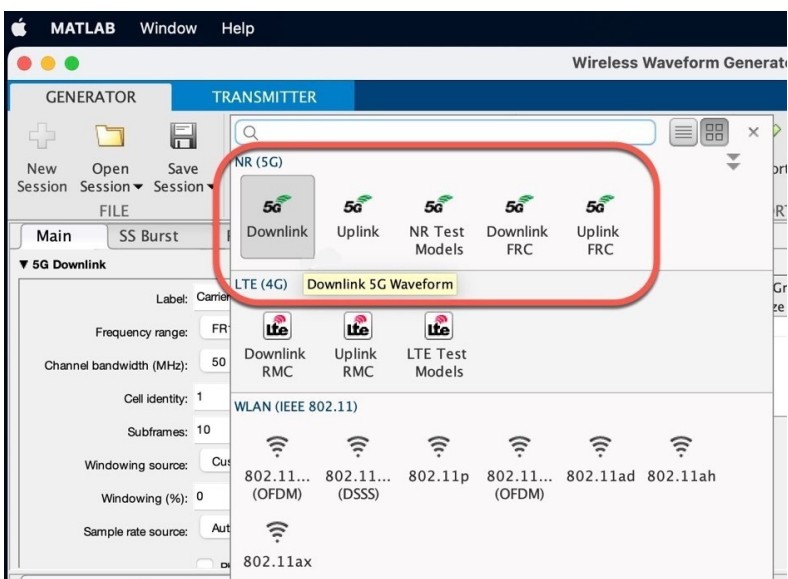

**Figure 18.** MATLAB provides various types of 5G signal simulations.

## 7. System Model and Decision Fusion Rules

### 7.1. Brief Description of the Problem

As shown in Figure 19, a total of *N* sensors are randomly distributed in the region of interest (ROI), where the ROI is a square area with area $b^2$. The *N* is a random variable that obeys a Poisson distribution, as Formula (4)

$$p(N) \ = \ \frac{\lambda^N e^{-\lambda}}{N!}, \ N \ = \ 0, \cdots, \infty \tag{4}$$

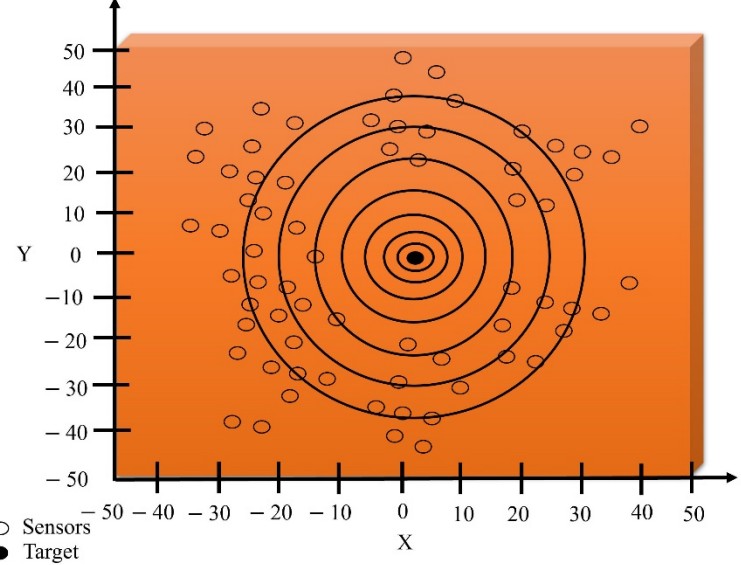

○ Sensors
● Target

**Figure 19.** Signal energy contours for targets located in the sensor area.

The locations of the sensors are unknown to the wireless sensor network, but they obey a common distribution in the ROI assuming they are independent and identically distributed (*i. d. d*), as Formula (5)

$$f(x_i, \ y_i) \ = \ \begin{cases} \frac{1}{b^2}, & -\frac{b}{2} \ll x_i, \ y_i \ll \frac{b}{2} \\ 0, & else \end{cases} \tag{5}$$

where $i = 1, 2, \ldots, N$, $(x_i, y_i)$ are the coordinates of sensor node $i$. The noise of the local sensor is independent and identically distributed, and all obey the standard Gaussian distribution, that is, the mean is 0, and the variance is 1, as Formula (6)

$$n_i \sim N\,(0, 1), i = 1, \ldots, N \tag{6}$$

For a local sensor $i$, the binary hypothesis testing problem is shown, as Formulas (7) and (8)

$$H_1: s_i = a_i + n_i \tag{7}$$

$$H_0: s_i = n_i \tag{8}$$

Among them, $s_i$ is the signal obtained at sensor $i$, and $a_i$ is the measured amplitude of the signal transmitted by the target received at sensor $i$. Using the isotropic signal energy attenuation model, the result can get following Formula (9)

$$a_i^2 = \frac{P_0}{1 + \alpha d_i^n} \tag{9}$$

where $P_0$ is the power of the signal emitted by the target at distance 0, and $d_i$ is the distance between the target and local sensor $i$, as Formula (10)

$$d_i = \sqrt{(x_i - x_t)^2 + (y_i - y_t)^2} \tag{10}$$

where $(x_t, y_t)$ are the coordinates of the target. It is further assumed that the location of the target obeys a uniform distribution on the ROI. The $n$ is the signal attenuation index, and its value is between 2 and 3. The $\alpha$ is a variable parameter, the larger $\alpha$ means the faster the signal attenuates. Such signal attenuation models can be easily extended to problems in three-dimensional space. The difference between them is that the denominator used in Formula (9) is $1 + \alpha d_i^n$ instead of $d_i^n$. This guarantees that the model is reasonable even when the distance $d_i$ is close to 0 or even equal to 0. When $d_i$ is large ($\alpha d_i^n \gg 1$), the difference between the two models can be ignored.

The type of passive sensor is not specified here, and the energy decay model used here is also generic. For example, in radar or wireless communication systems, for isotropically radiated electromagnetic waves propagating in free space, the power is inversely proportional to the square of the distance from the transmitter. Similarly, when a simple sound source emits a spherical sound wave outward in the air, its sound wave intensity is also inversely proportional to the square of the distance. Since the noise has unit variance, it is clear that the signal-to-noise ratio ($SNR$) of the local sensor $i$ is shown following Formula (11)

$$SNR_i = a_i^2 = \frac{P_0}{1 + \alpha d_i^n} \tag{11}$$

The $SNR$ at zero distance is defined as following Formula (12)

$$SNR_0 = 10 log_{10} P_0 \tag{12}$$

It is assumed that all local sensors use the same threshold $\tau$ to make judgments under the condition of Gaussian noise. The false alarm rate and detection probability of the local sensor can be obtained as Formulas (13) and (14)

$$p_{fa} = \int_\tau^\infty \frac{1}{\sqrt{2\pi}} e^{\frac{-t^2}{2}} dt = Q(\tau) \tag{13}$$

$$p_{d_i} = \int_\tau^\infty \frac{1}{\sqrt{2\pi}} e^{\frac{-(t - a_i)^2}{2}} dt = Q(\tau - a_i) \tag{14}$$

where $Q(\cdot)$ is the complementary distribution function of the standard Gaussian distribution, as Formula (15)

$$Q(x) = \int_{\tau}^{\infty} \frac{1}{\sqrt{2\pi}} e^{\frac{-t^2}{2}} dt \tag{15}$$

It is assumed that the ROI is large, and that the signal energy decays rapidly. Therefore, in only a relatively small part of the ROI, that is, the area around the target, the received signal energy will be significantly greater than zero. Without loss of generality, ignore the boundary effects of the ROI and assume that the target is in the center of the ROI. Therefore, at a given moment, only a fraction of the sensors can detect the target. In order to reduce the communication cost and energy consumption, the local sensors will only send data to the fusion center when the threshold $r$ is exceeded.

*7.2. Decision Fusion Rules*

Denote the binary data obtained from the local sensor by $I_i = \{0,1\}$ ($i = 1, \ldots, N$). If a target is detected, the value of $I_i$ is 1; if no target is detected, the value of $I_i$ is 0. The Chair-Varshney fusion rule is an optimal decision fusion rule. This system tests the threshold of the following data, as following Formula (16)

$$
\begin{aligned}
\Lambda_0 &= \sum_{i=1}^{N} \left[ I_i \log \frac{p_{d_i}}{p_{fa_i}} + (1 - I_i) \log \frac{p_{d_i}}{p_{fa_i}} \right] \\
&= \sum_{i=1}^{N} I_i \log \frac{p_{d_i}(1-p_{fa_i})}{p_{fa_i}(1-p_{d_i})} + \sum_{i=1}^{n} \log \frac{1-p_{d_i}}{p_{fa_i}}
\end{aligned}
\tag{16}
$$

This fusion data is equivalent to a weighted sum of all detections ("1") received by the fusion center. If the detection performance of a certain sensor is better, that is, $p_{d_i}$ is higher and $p_{fa_i}$ is lower, then its decision will get higher weight, and its weight can be expressed as Formula (17)

$$\log \frac{p_{d_i}(1-p_{fa_i})}{p_{fa_i}(1-p_{d_i})} \tag{17}$$

According to Equation (13), once the threshold $r$ is known, the false alarm rate of each sensor is also known. However, it is very difficult to find $p_{d_i}$ for each sensor. According to Equation (14), $p_{d_i}$ is determined by the distance from each sensor to the target and the amplitude of the target signal. To make matters worse, the total number $N$ of sensors cannot be known since the fusion center can obtain data from a sensor only when the signal received by the sensor exceeds the threshold $r$. Another strategy is that each sensor sends raw data $s_i$ to the fusion center, which then makes decisions based on these raw data. However, transmitting raw data can be costly, especially for wireless sensor networks with limited energy and bandwidth. However, ust transferring binary data to the fusion center is acceptable. If $p_{d_i}$ is unknown, the fusion center can only treat the detection results from each sensor indiscriminately. An intuitive choice is to count the number of occurrences of the detection result "1", because the result "1" sent by a single sensor is almost useless to the fusion center.

## 8. Fifth Generation (5G) Communication Module Connection

Our research uses some 5G communication modules such as SIM8200EA-M2 and RM500Q-GL for sensed data transmission. The SIM8200EA-M2 core module, based on the Qualcomm Snapdragon X55 platform, has multi-mode and multi-band support. For the actual wiring of the SIM8200EA-M2, refer to Figure 20. The SIM8200EA-M2 5G communication module supports 5G/4G/3G multi-mode Internet access, and can make calls, send text messages, offer cloud platform communication, and so on.

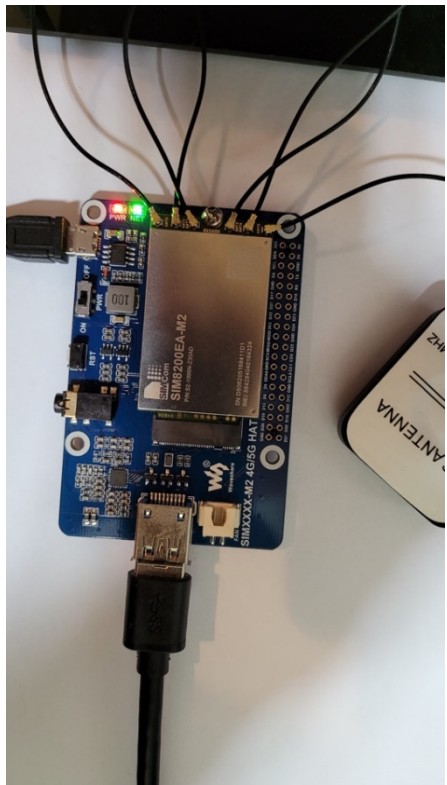

**Figure 20.** Wiring of SIM8200EA-M2.

Another 5G communication module in this paper is RM500Q-GL, a product developed by Quectel. The RG500Q is a series of 5G sub-6 GHz LGA modules optimized for IoT and M2M applications. Figure 21 shows the actual operation and usage of RM500Q-GL.

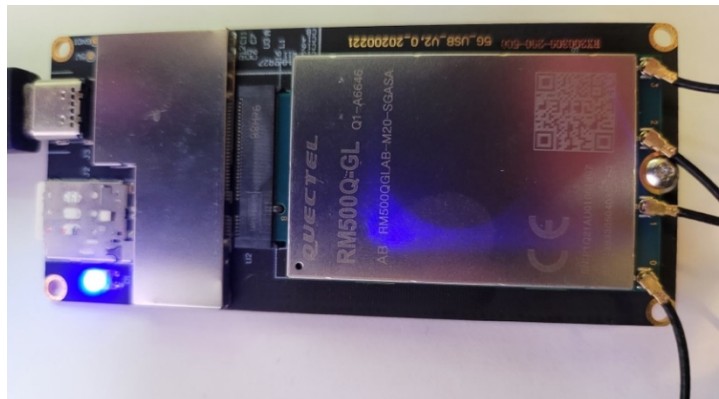

**Figure 21.** Actual usage of RM500Q-GL.

The 5G communication modules discussed in this paper were all connected using standard 5G-dedicated antenna modules, as shown in Figure 22.

After completion of the wiring and installation of drivers, the device administrator of the computer system displayed the Qualcomm Snapdragon X55 5G item in the network device, as shown in Figure 23. This situation indicates that the 5G communication module was successfully installed, and that the system could then use 5G communications to transmit data.

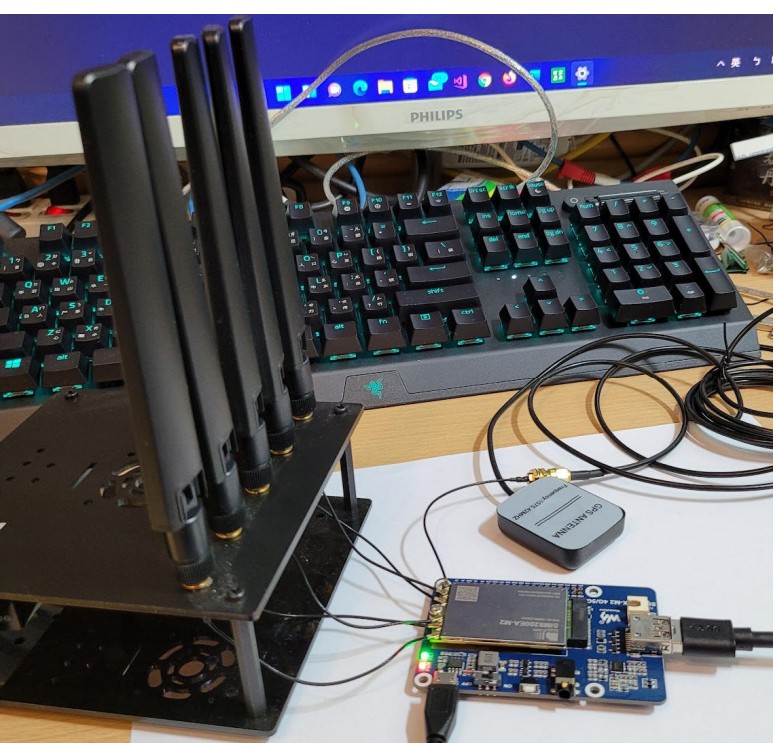

**Figure 22.** The 5G-dedicated antenna connection used in our experiments.

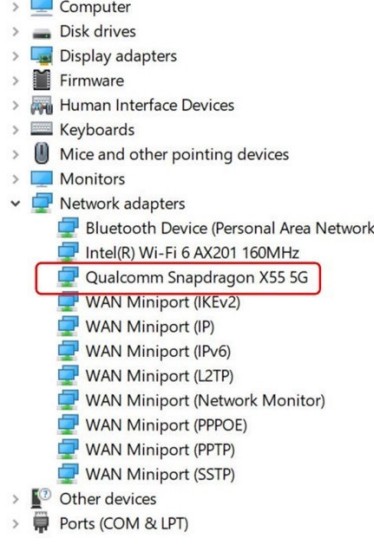

**Figure 23.** Qualcomm Snapdragon X55 5G project.

Our research uses the Speedtest speed measurement software to test the speed of the 5G network. After many repeated tests, the 5G speed reached a certain level. Please refer to the speed measurement screen shown in Figure 24.

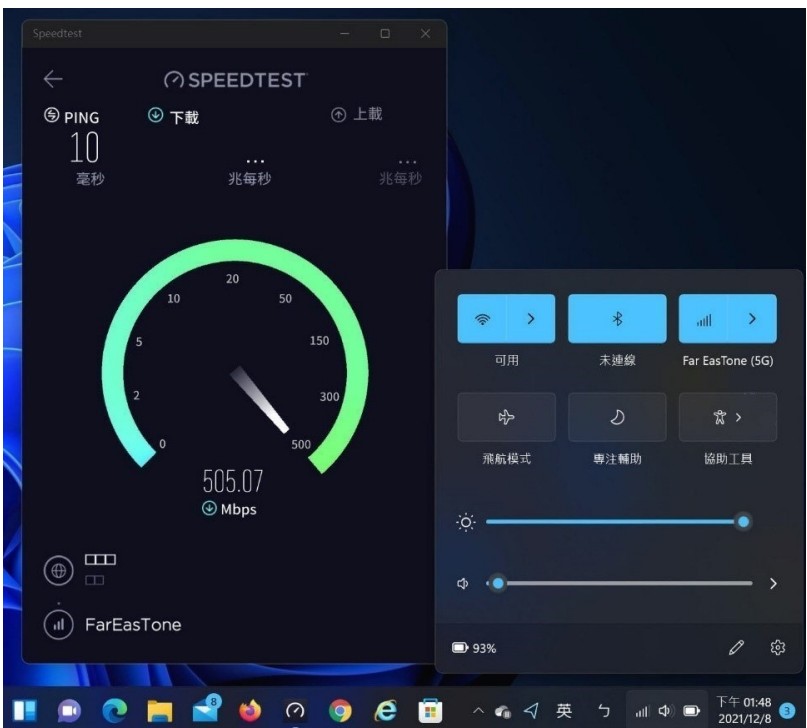

**Figure 24.** 5G test speed scenarios.

## 9. Conclusions

In this paper, our approach uses 5G communications to transmit remote sensing signals. The advantages of using a 5G system include the system's ability to ensure the safe transmission of the data. Moreover, the proposed method can effectively improve the transmission efficiency of sensed data, and the system can transmit pictures and audio-visual data. Our research uses MATLAB to simulate the transmission analysis of 5G signals and the setting of other communication parameters. At the same time, our team also discussed the application scenarios of network slicing technology, as different network slicing technologies will be used when transmitting various data. When the remote sensing signal is transmitted to the embedded hardware module, the system processes the signals through use of an AI algorithm. The system analyzes and classifies the sensed data through various AI algorithms, and then transmits the results to a cloud database system through 5G communications, allowing real-time data display and visualization. At the end of this paper, an experiment in which 5G communication hardware modules were connected was presented, and our team conducted data transmission and network speed tests. From the results of the 5G test speed in this paper, it is concluded that the efficiency of 5G data transmission will facilitate future technological developments, which was also the focus of this paper.

**Funding:** The author received no specific funding for this study.

**Data Availability Statement:** Data sharing not applicable to this article as no datasets were generated or analyzed during the current study.

**Acknowledgments:** This research was supported by the Department of Information Technology, Takming University of Science and Technology. The authors would like to thank the Takming University of Science and Technology, for financially supporting this research.

**Conflicts of Interest:** The authors declare that they have no conflict of interest to report regarding the present study.

## Appendix A

Fifth generation New Radio (5G NR) is a new radio access technology (RAT), developed by 3GPP for the fifth-generation mobile communication network. The 5G NR is a global standard for 5G network air interfaces. The Third Generation Partnership Program (3GPP)'s 38 series of specifications define the technical details for NR. Their research on NR began in 2015, with the first specification release coming out in late 2017. At that time, the 3GPP standardization journey was continuing, and the industry had already begun to implement those draft standard-compliant infrastructures, with the first large-scale commercial deployments of 5G NR expected to take place in 2019.

The 5G NR bands are generally divided into two frequency ranges:

1.  Frequency range 1 (FR1), including the frequency band below 6 GHz (sub-6 GHz), which currently ranges from 410 MHz to 7125 MHz.
2.  Frequency range 2 (FR2), including frequency bands in the mmWave range (24.25–52.6 GHz, to be exact).

FR2 has a smaller range, but has more available frequency bands than FR1. Please refer to the settings of FR1 and FR2 shown in Figure A1.

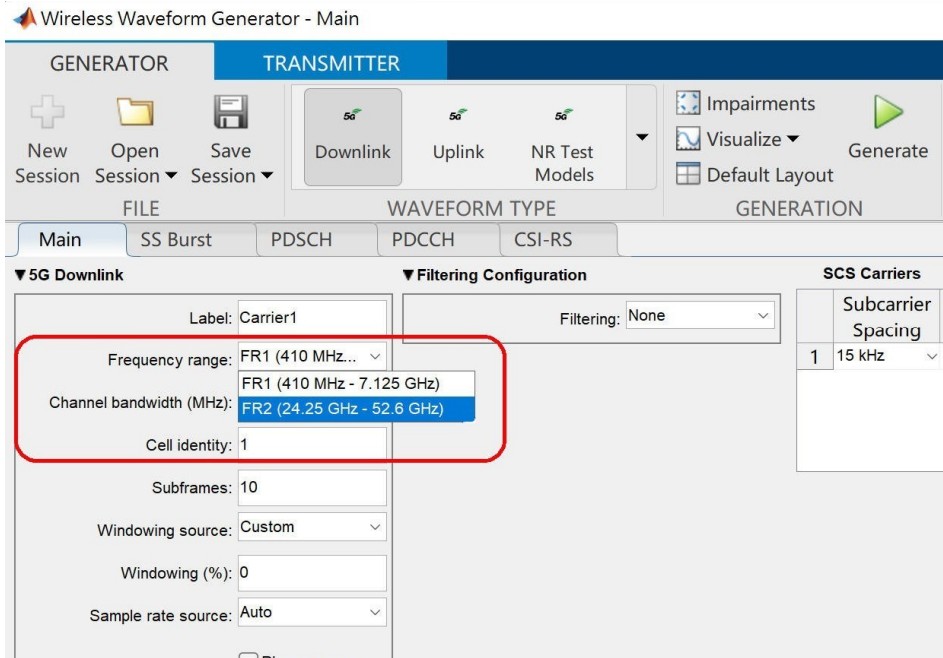

**Figure A1.** Setting the bandwidths of FR1 and FR2.

The initial 5G NR deployment will rely on the existing long-term evolution (LTE) 4G infrastructure in a non-standalone (NSA) mode, until the standalone (SA) mode that works in conjunction with the 5G core network matures. The non-standalone (NSA) mode of 5G NR refers to an option for 5G NR deployments in which control functions rely on the control plane of the existing LTE network, whereas 5G NR focuses entirely on the user plane. The advantage of this approach is that it speeds up the progress of 5G commercialization. However, some operators and device manufacturers have criticized this, arguing that the early introduction of 5G NR NSA will hinder the introduction of independent networking (i.e., SA) mode networks. The independent networking (i.e., SA) mode of 5G NR refers to the use of 5G base stations for both signaling and data transmission. It uses the new 5G packet-switched core network architecture, instead of the 4G core network evolved packet core (EPC). The 5G network deployment in SA will be completely independent of the 4G network. It is expected to have a lower cost, higher efficiency, and will help to develop new use-cases.

The MATLAB 5G Waveform Generator application provides a UI to manage numerous configuration parameters. Within the app, researchers can select waveform types, specify parameters, and generate and export waveforms. The application also enables operators to interact with test and measurement equipment.

Figure A2 shows the configuration of two physical down-link shared channels (PDSCH). The first PDSCH spans all slots and uses physical resource blocks (PRBs) 0 to 100. The second PDSCH is active in slots 0 to 2 and 4 to 6 and uses PRBs 200 to 250.

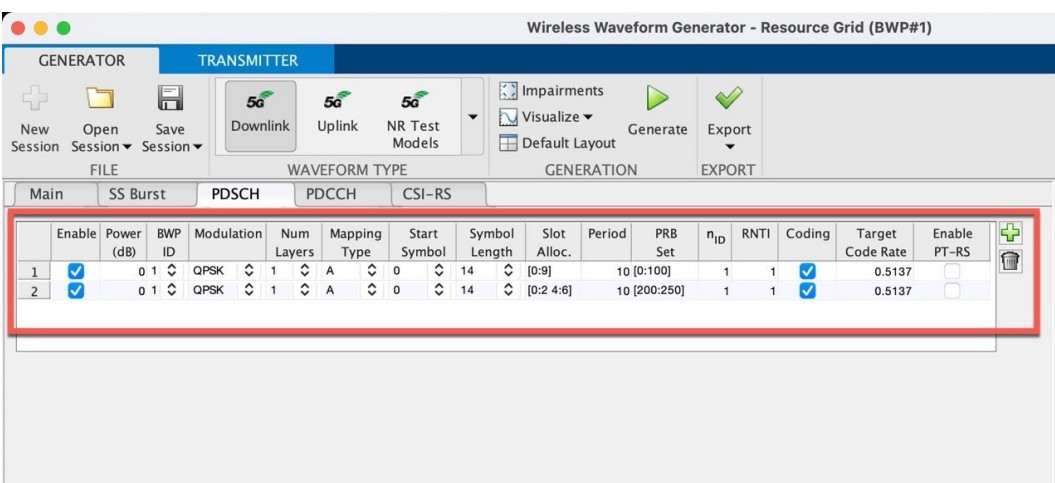

**Figure A2.** Parameter settings for the PDSCHs.

According to the above parameter settings, the system generated a 5G signal. Figure A3 shows the spectral analysis of the signal.

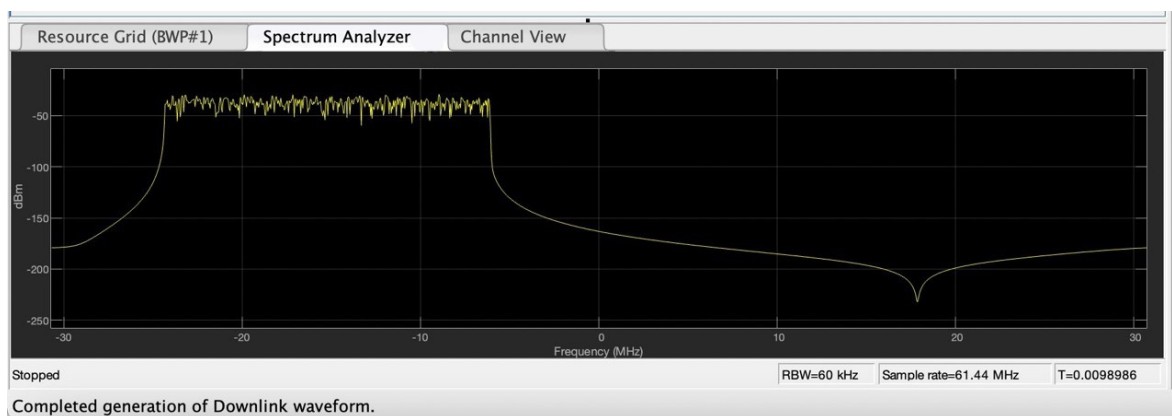

**Figure A3.** The 5G signal set by the developer.

This study used MATLAB code to write and generate 5G waveforms. For example, our code uses nrDLCarrierConfig to configure and generate 5G downstream waveforms. The code generated for the downlink waveform, using the export to MATLAB script option in the application, was also configured using nrDLCarrierConfig (see Figure A4).

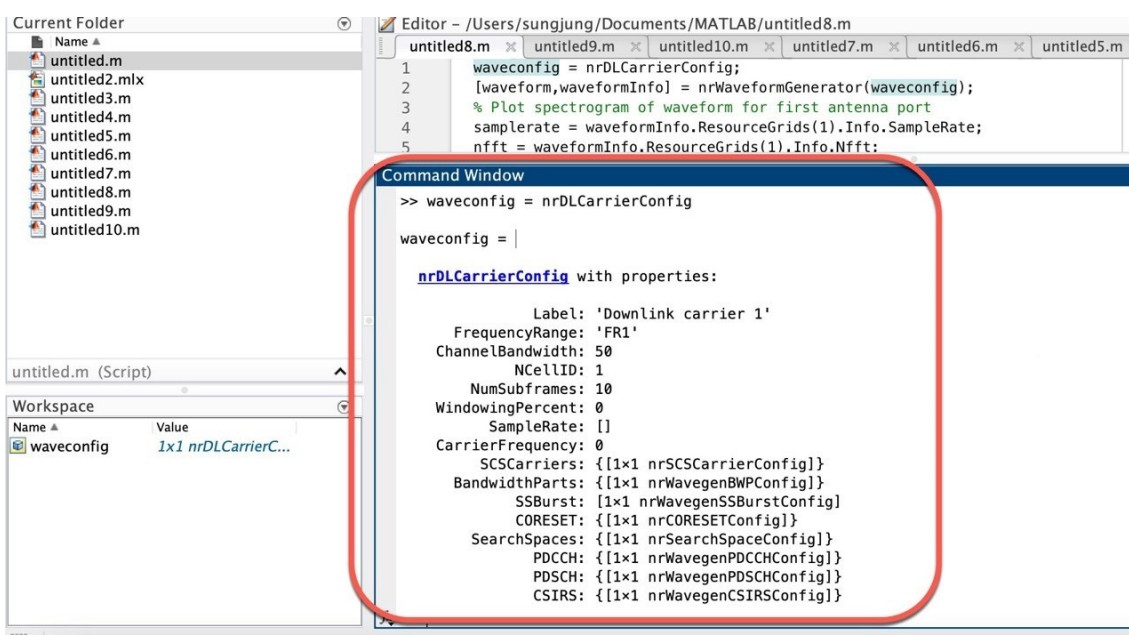

**Figure A4.** Code used to generate a 5G signal.

A spectrogram was drawn in MATLAB, in order to visualize the signal in the frequency domain. The waveform includes the fully allocated PDSCH, the physical downlink control channel (PDCCH), and signal synchronization (SS) bursts; please refer to Figure A5.

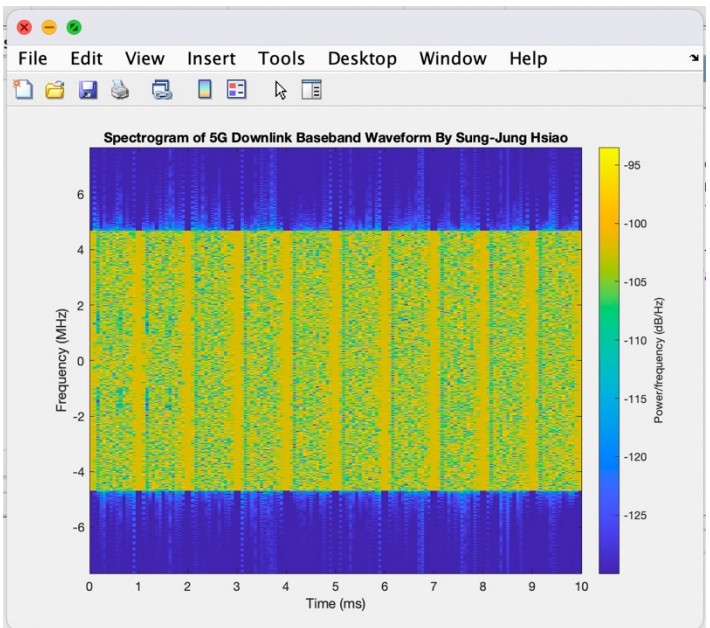

**Figure A5.** Fifth generation (5G) down-link baseband waveform spectrum.

The proposed approach changed the PDSCH allocation to span PRBs 0 to 10. The system then generated a waveform and drew a spectrogram, which is shown in Figure A6.

Multiple instances of physical channels and signals can also be defined. Our approach created a second instance of the PDSCH configuration object and set the allocation to span PRB 40 to 50 and OFDM symbols 2 to 10. Our research then assigned the second PDSCH configuration to the waveform configuration, and the system generated the waveform shown in Figure A7.

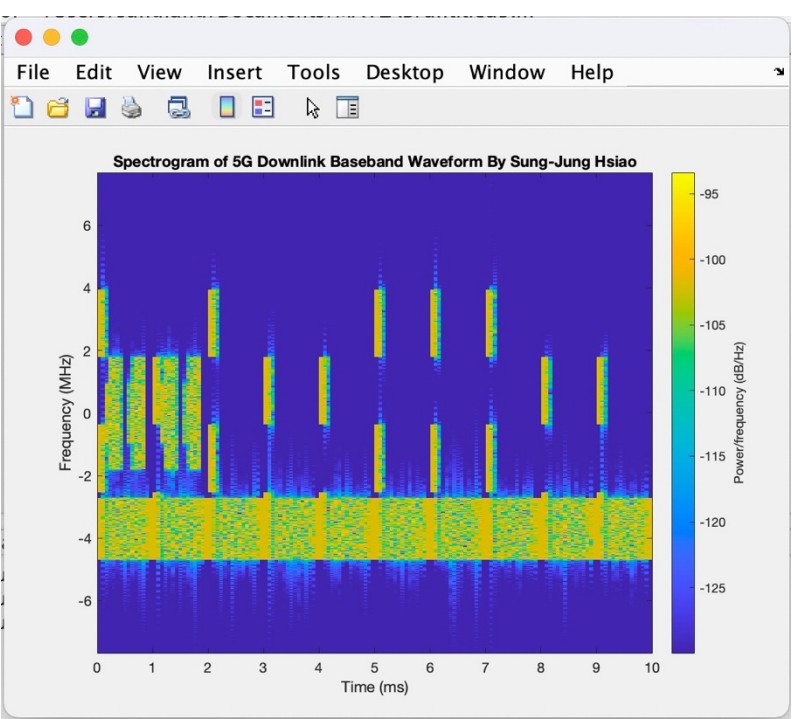

**Figure A6.** Spectrogram of a 5G down-link baseband waveform with a PRB ranging from 0 to 10.

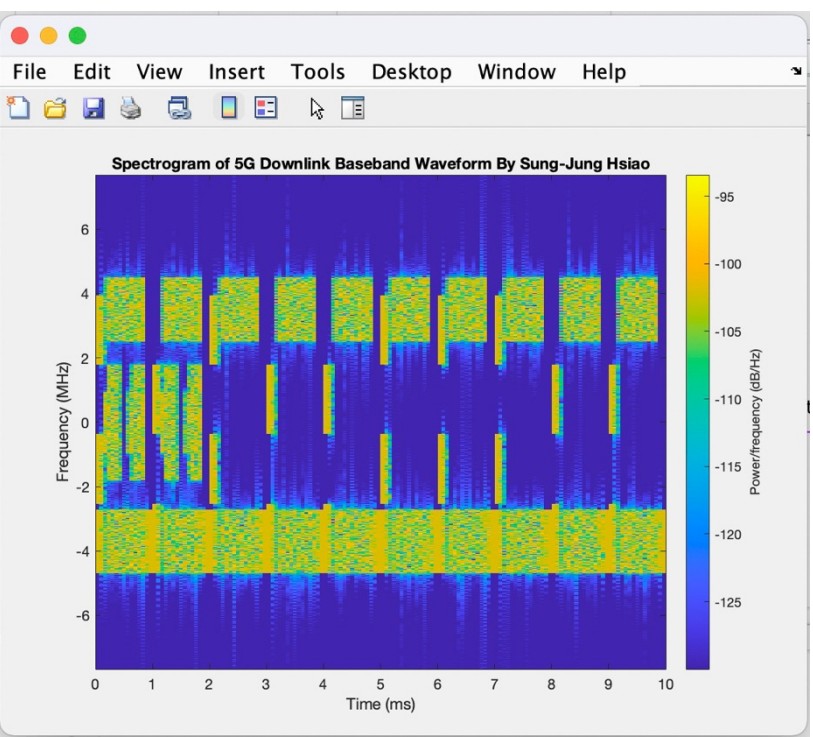

**Figure A7.** PRB values 40 to 50 and OFDM symbols 2 to 10.

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
