# Peer review of "Employing a Wireless Sensing Network for AIoT Based on a 5G Approach"

_electronics, doi:10.3390/electronics11050827_

Round 1
Reviewer 1 Report
In this review article, the author reviews the advantages of using a 5G system to transmit remote sensing signals. The summary of 5G systems is great and if the author includes the current challenges and required improvements of 5G systems, that will improve the quality of the manuscript a lot. Also, the last part “the experiment using 5G communication hardware modules” should not be a part of a review article. This part is more like a technical report. I suggest moving this part to supplementary information.
- In the title, the author should provide the full name of ‘AIoT’.
- In the Abstract, the author should give the full name of ‘NB-IoT’. Since there are a lot of terms in the manuscript, especially in figures, it is recommended that the author make a table with the terms and associated full names.
- The reviewer recommends the author should also mention the cost of 5G, compared with previous technologies. I assume that the requirement of high-density of cells increases the cost dramatically.
- The author reviewed the 5G standardization timeline from 2013 to 2020. Two comments: [1] Now already 2022. The author should update the status in 2021. [2] There is no outlook. The author should outlook what the 5G technology should be and will be starting from now.
- I am not sure why the author adds the part “7. The implementation of 5G network technology”. This part is more like a technical manual instead of a review. I suggest putting this part into supplementary information.
Reviewer 2 Report
In this paper, the author showed that the 5G mobile radio system can be used to transmit data acquired by sensors. Unfortunately, almost all of the paper describes the 5G system, exposing concepts that are very well known in the technical literature, and without introducing any degree of novelty. However, it is a novelty that is difficult to introduce in a system that is now standardized and operational.
Finally, in a few words at the end of the paper, the author describes the Matlab generation of the 5G signal and the possibility of using commercial boards to transmit data. The paper does not present any novelty that could justify its publication, nor of scientific interest. No, this article has no technical or scientific relevance, and cannot be taken into consideration for publication.
Round 2
Reviewer 1 Report
The manuscript has improved a lot. Thanks for the author's effort and time. I am looking forward to its publication!
Reviewer 2 Report
The improved version of paper can be recommended for publishing.